

# Barriers to care and the need for dental educational materials for the Lowe syndrome community: a survey of dentists

Adam Lowenstein[1], Matthew Finkelman[2], Jay Dalal[3], Crystal Smith[3], Glory Ogunyinka[3], David Tesini[4] and Carlos Fernando Mourão[1]

[1] Department of Basic and Clinical Translational Sciences, Tufts University, Boston, MA, United States
[2] Department of Public Health and Community Service, Tufts University, Boston, MA, United States
[3] Undergraduate Dental Student, Tufts University, Boston, MA, United States
[4] Pediatric Dentistry, Tufts University, Boston, MA, United States

## ABSTRACT

**Background:** This study aimed to assess dentists' experience in treating individuals with Lowe syndrome (LS), reasons they may be unable to provide dental care for individuals with LS, and perceptions of the need for educational materials tailored to the LS community regarding the dental setting.

**Methods:** A link to an electronic Qualtrics survey addressing these topics was emailed to the Tufts University School of Dental Medicine Alumni Network listserv.

**Results:** Data from 73 respondents were analyzed. Of the 57 who answered the question about having treated a patient with LS, three (5.3%) responded affirmatively. Among the 61 who answered why they might not be able to treat an individual with LS, the most common reasons were lack of experience treating children with special needs and not accepting medical assistance such as Medicaid/Medicare (both 31.1%). Of the 58 who responded to the item regarding the need for educational materials to support patients with LS in the dental setting, 47 (81.0%) agreed or strongly agreed.

**Conclusion:** Substantial barriers to dental care exist for individuals with LS. Educational materials about the dental setting should be developed for the LS community.

Corresponding author
Carlos Fernando Mourão,
carlos.mourao@tufts.edu

## INTRODUCTION

Lowe syndrome (LS), also known as oculocerebrorenal syndrome (OCRL), is a rare disorder. It is a rare disorder typically characterized by abnormalities in the eyes, kidneys, central nervous system, and/or brain (*Bockenhauer et al., 2008*; *Brooks & Ahmad, 2009*; *Coon et al., 2012*; *Loi, 2006*; *Ramirez et al., 2012*). Children with LS have congenital cataracts (*Bockenhauer et al., 2008*; *Brooks & Ahmad, 2009*; *Coon et al., 2012*; *Loi, 2006*; *Ramirez et al., 2012*; *Schurman & Scheinman, 2009*), with glaucoma also present in approximately 50% of individuals with the condition (*National Library of Medicine, 2022*). Associated renal problems include proteinuria, generalized aminoaciduria, and acidosis,

while problems related to the central nervous system include psychomotor impairment and hypotonia (*Ruellas et al., 2011*, *2008*); delayed intellectual development is also common (*National Library of Medicine, 2022*). Other manifestations include behavioral issues, seizures, breathing and feeding difficulties, rickets, scoliosis, deviations from the norm in height and weight, and shortened life span (*Loi, 2006*; *National Library of Medicine, 2022*; *Ruellas et al., 2011*, *2008*; *Holtgrewe & Kalen, 1986*; *Richards et al., 1965*).

LS is caused by a mutation of the oculocerebrorenal gene, OCRL-1, localized to Xq24-q26 (*Brooks & Ahmad, 2009*). An X-linked, recessive disorder (*Brooks & Ahmad, 2009*), it occurs nearly exclusively in males (*National Library of Medicine, 2022*; *Lowenstein et al., 2023*). Its prevalence has been estimated broadly as between one and 10 per 1,000,000 people (*Loi, 2006*), and more specifically as approximately one per 500,000 people. Globally, it is estimated similarly to the United States, but dental care access for rare diseases varies due to differing healthcare systems and training standards (*Brooks & Ahmad, 2009*; *National Library of Medicine, 2022*).

Individuals with LS often experience increased dental problems. Some authors have divided these problems into seven overlapping categories, including difficulties with teeth (such as crowding, decay, and misalignment including a double row of teeth in some individuals, among other abnormalities), gingiva, extractions, dental cysts, the need for general anesthesia for dental procedures, dental surgery, and braces/orthodontic devices (which have been reported to be ineffective in most individuals with LS) (*Knight & Scientific Advisory Board, 2010*). Case reports in the dental literature have found delayed eruption, generalized tooth mobility, enlarged pulp chambers, enamel hypoplasia, dysplastic dentin formation, eruption cysts, hematomas, tooth staining from iron medication (prescribed to treat anemia), incompetent lips, and taurodontism (*Brooks & Ahmad, 2009*; *Ruellas et al., 2011*, *2008*; *Batirbaygil & Turgut, 1999*; *Harrison, Odell & Sheehy, 1999*; *Roberts et al., 1994*; *Rodrigues Santos et al., 2007*; *Thomas & Grimm, 1994*). Despite the prevalence of dental issues among individuals with LS, there is a relative lack of research on the topic. A 1991 study reported that subjects with the disorder were more likely than the general population to experience many of the problems listed above, including misalignment, extractions, and general anesthesia appointments (*McSpadden, Dolinsky & Schroerlucke, 1991*). A 1999 study also found a high prevalence of misalignment, extractions, gingival bleeding, dental restorations, dental cysts, and behavioral issues at the dental office (*Lowenstein et al., 2023*). More recently, a 2023 survey study found that individuals with LS were not only more likely to have more reported deleterious dental conditions (tooth misalignment, difficulty upon mastication, halitosis, and intraoral lesions) and fewer healthy dental hygiene practices (brushing at least twice per day, flossing, brushing themselves, and being accepting of brushing and flossing) than healthy individuals, but also greater difficulty in accessing dental care. Specifically, 15% of parents/guardians of individuals with LS reported that a dentist was unable to provide treatment due to having an office that was not properly equipped, and 21% reported that a dentist was unable to provide treatment because they did not have experience treating those with special needs. Perhaps most alarmingly, only 13% reported that it was "very easy" to locate a dentist for the individual with LS, while 23%, 23%, 20%, and 20% reported

that it was "somewhat easy," "neither easy nor hard," "somewhat difficult," and "very difficult," respectively (*Lowenstein et al., 2023*).

Given the above-mentioned findings, improving access to dental care for individuals with LS would constitute a great stride forward for the oral health of this population. A fundamental step in the process is to understand dentists' experience and current limitations in treating individuals with LS, as well as their perceptions of how dental knowledge can be best disseminated to individuals in the LS community (*e.g.*, individuals with LS and their parents/caregivers), so that future interventions can be designed accordingly. By examining the dentist's perspective on LS care barriers, we are filling a critical gap in the literature where only patient/caregiver perspectives have been previously documented. Therefore, the primary aim of this study was to assess dentists' experience in treating individuals with LS, reasons why they would not be able to provide dental care for individuals with LS, and perceptions of the need for educational materials about the dental setting that are tailored to the LS community. In addition, information about which dentist-level factors are associated with an inability to treat individuals with LS and their perceptions of the need for dental educational materials for the LS community would shed light on specific areas to target in future interventions. Therefore, the secondary aim was to evaluate associations between dentist-level variables and (1) reasons for not being able to treat a person with LS and (2) perceptions of the need for dental educational materials for the LS community. The authors hypothesize that most dentists would have limited or no experience treating LS patients. In addition, dentists would identify multiple significant barriers to providing care, and dentists would strongly support the development of educational materials.

## MATERIALS AND METHODS

This cross-sectional survey study was conducted in accordance with ethical guidelines and evaluated by the Institutional Review Board (IRB) of Tufts University School of Dental Medicine (IRB Protocol Number: 00004167—Exempt Determination). All participants were informed of the study's purpose, procedures, and their rights, and informed consent was obtained online *via* Qualtrics. The IRB ensured that the study complied with ethical standards to protect the confidentiality, welfare, and rights of all participants involved.

A 28-item survey for dentists was developed for this research, including items on demographics; dental specialty, years of dental experience, and current volume of clinical work; experience in treating patients with LS; reasons why the dentist might not be able to treat an individual with LS; and the need for dental educational materials for the LS community. The survey, after expert consultation with the study team, was pre-tested for content validity and face validity. Regarding the evaluation of content validity, three dentists at Tufts University School of Dental Medicine (TUSDM) were provided the survey and were asked to rate each item's level of importance on a five-point Likert scale (1 = very important, 2 = important, 3 = moderately important, 4 = of little importance, or 5 = not important). In addition, they were asked to rate whether each item should be included in the survey (0 = no, 1 = unsure, or 2 = yes). Regarding the evaluation of face validity, three

dentists at TUSDM who were not involved in content validation reviewed the survey to assess whether the items were easily understood, simple, useful, and necessary.

A link to an electronic Qualtrics (Qualtrics, Provo, UT, USA) survey was emailed to the listserv of TUSDM's alumni network. Inclusion criteria were TUSDM's alumni who reported at the start of the survey that they were currently in the United States and were at least 18 years old. Exclusion criteria were the following: non-TUSDM graduates, not currently practicing in the United States, under 18 years old, and did not provide informed consent. The survey was open from September 26, 2023 to December 12, 2023. A reminder email was sent after 4 weeks.

Descriptive statistics (frequencies and percentages) were calculated. For binary outcome variables, statistical significance was evaluated using the chi-square test (or Fisher's exact test in the case of small expected cell counts). For ordinal outcome variables, statistical significance was evaluated using the Mann-Whitney U test. The significance level was set at $\alpha = 0.05$. SPSS v. 28 (IBM Corp., Armonk, NY, USA) was used in the analysis.

## RESULTS

Seventy-nine initial responses to the survey were obtained (considering that Tufts University School of Dental Medicine's listserv includes approximately 5,500 email addresses, the initial response rate was approximately 1.4%). Data from six of these subjects were not included in the statistical analysis (three subjects responded that they did not consent to the survey; one did not answer the item about consenting to the survey; one responded that they were not currently in the United States; and one did not answer the item about currently being in the United States), yielding a convivence sample size of 73. The rationale for choosing the single institution is as follows: Tufts University School of Dental Medicine has a large, geographically diverse alumni network spanning multiple decades of graduates, the institution has an established curriculum in special care dentistry, and access to the alumni network was readily available for research purposes. The curriculum includes didactic coursework on developmental disabilities, clinical rotations, and case-based learning. As some of these 73 subjects provided responses to some items and not others, sample sizes varied across the different survey items.

Table 1 presents reported characteristics of the study sample. Based on the observed distributions of professional characteristics among the sample, the following categories were created for subsequent comparative analysis: general dentists *vs.* specialists; 0–20 years of experience *vs.* 21+ years of experience; currently seeing patients 0–3 days per week *vs.* 4+ days per week; and currently seeing 0–40 patients per week *vs.* 41+ patients per week.

Table 2 shows subjects' reported experience (or lack thereof) in having treated a patient with LS, potential reasons for their being unable to treat an individual with LS, and their perceived need for dental educational materials for the LS community. Of the 57 subjects who responded to the item inquiring about having ever treated a patient with LS, three (5.3%) answered positively. Among the 61 subjects who responded to the item inquiring about reasons why they might not be able to treat an individual with LS, 46 (75.4%) reported at least one reason; these specific data are summarized in the text but not presented in tabular form. The most common reported reasons were that they do not have

**Table 1 Reported characteristics of the study sample.**

| Variable | Category | n | % |
|---|---|---|---|
| Race | White | 58 | 79.5 |
| | Black or African American | 1 | 1.4 |
| | American Indian and Alaska Native | 1 | 1.4 |
| | Asian | 7 | 9.6 |
| | Native Hawaiian or Pacific Islander | 0 | 0.0 |
| | Other | 6 | 8.2 |
| Dental specialty | General dentist | 42 | 60.9 |
| | Endodontist | 2 | 2.9 |
| | Orthodontist | 5 | 7.2 |
| | Oral maxillofacial surgeon | 3 | 4.3 |
| | Oral pathologist | 0 | 0.0 |
| | Pediatric dentist | 8 | 11.6 |
| | Periodontist | 7 | 10.1 |
| | Prosthodontist | 2 | 2.9 |
| Years of dental experience (Excluding student/Resident experience) | 0–5 | 5 | 7.4 |
| | 6–10 | 8 | 11.8 |
| | 11–15 | 5 | 7.4 |
| | 16–20 | 4 | 5.9 |
| | 21+ | 46 | 67.6 |
| Number of days per week currently seeing patients | 0 | 16 | 22.9 |
| | 1 | 3 | 4.3 |
| | 2 | 3 | 4.3 |
| | 3 | 6 | 8.6 |
| | 4 | 21 | 30.0 |
| | 5 | 20 | 28.6 |
| | 6 | 1 | 1.4 |
| | 7 | 0 | 0.0 |
| Number of patients currently seen per week | 0 | 16 | 24.6 |
| | 1–20 | 5 | 7.7 |
| | 21–30 | 9 | 13.8 |
| | 31–40 | 2 | 3.1 |
| | 41–50 | 9 | 13.8 |
| | 51+ | 24 | 36.9 |

experience treating children with special needs and that they do not accept medical assistance such as Medicaid/Medicare (both 31.1%). Of the subjects who replied "Other" to this item and provided their own reason, the most common answer (provided by five subjects) was that they had not previously heard of LS. Among the 58 subjects who responded to the item asking for their level of agreement that there is a need for more educational materials to help patients with LS in the dental setting, 47 (81.0%) agreed or strongly agreed, while none disagreed or strongly disagreed. Of the 56 subjects who responded to the item asking about the types of educational materials that would be helpful

**Table 2 Reported experience treating patients with Lowe syndrome (LS).**

| Variable | | Category | *n* | % |
|---|---|---|---|---|
| Treated a patient with Lowe syndrome | | Yes | 3 | 5.3 |
| | | No | 54 | 94.7 |
| Reasons why subject might not be able to treat an individual with Lowe syndrome | Do not have experience treating children with special needs | Yes | 19 | 31.1 |
| | | No | 42 | 68.9 |
| | Do not accept medical assistance (*e.g.*, Medicaid/Medicare) | Yes | 19 | 31.1 |
| | | No | 42 | 68.9 |
| | Requires a multidisciplinary approach | Yes | 11 | 18.0 |
| | | No | 50 | 82.0 |
| | Dental office not properly equipped | Yes | 11 | 18.0 |
| | | No | 50 | 82.0 |
| | Other | Yes | 14 | 23.0 |
| | | No | 47 | 77.0 |
| Feel there is a need for more educational materials to help patients with Lowe syndrome in the dental setting | | Strongly agree | 25 | 43.1 |
| | | Agree | 22 | 37.9 |
| | | Neutral | 11 | 19.0 |
| | | Disagree | 0 | 0.0 |
| | | Strongly disagree | 0 | 0.0 |
| Feel the following types of educational materials would be helpful for individuals with Lowe syndrome | Website/Mobile application | Yes | 37 | 66.1 |
| | | No | 19 | 33.9 |
| | Introductory dental video | Yes | 41 | 73.2 |
| | | No | 15 | 26.8 |
| | Pamphlets with oral hygiene instructions | Yes | 40 | 71.4 |
| | | No | 16 | 28.6 |
| | Media channels (*e.g.*, TikTok/Instagram/YouTube) | Yes | 17 | 30.4 |
| | | No | 39 | 69.6 |
| | Podcasts featuring dental experts | Yes | 19 | 33.9 |
| | | No | 37 | 66.1 |
| | Focus groups | Yes | 16 | 28.6 |
| | | No | 40 | 71.4 |

for individuals with LS, the most frequently selected answers were an introductory dental video (73.2%), pamphlets with oral hygiene instructions (71.4%), and a website/mobile application (66.1%).

Table 3 presents associations between reported dentist-level variables and reasons for not being able to treat a person with Lowe syndrome. General dentists, subjects with 21+ years of dental experience, and subjects currently seeing 0–40 patients per week were significantly more likely to report a lack of experience treating children with special needs as a reason why they might not be able to treat an individual with Lowe syndrome ($p = 0.022$, $p < 0.001$, and $p = 0.003$, respectively). All other associations were not statistically significant.

Tables 4 and 5 show associations between reported dentist-level variables and perceptions of the need for dental educational materials for the LS community. Subjects
**Table 3 Associations between reported dentist-level variables and reasons for not being able to treat a person with Lowe syndrome[†].**

| | | | Dental specialty | | Years of dental experience | | Number of days per week currently seeing patients | | Number of patients currently seen per week | |
|---|---|---|---|---|---|---|---|---|---|---|
| | | | General dentist | Specialist | 0–20 | 21+ | 0–3 | 4+ | 0–40 | 41+ |
| Reasons why subject might not be able to treat an individual with Lowe syndrome | Do not have experience treating children with special needs | Yes | 15 (42.9) | 4 (15.4) | 0 (0.0) | 19 (47.5) | 10 (43.5) | 9 (23.7) | 14 (50.0) | 5 (15.2) |
| | | No | 20 (57.1) | 22 (84.6) | 20 (100.0) | 21 (52.5) | 13 (56.5) | 29 (76.3) | 14 (50.0) | 28 (84.8) |
| | | p | 0.022* | | <0.001* | | 0.106 | | 0.003* | |
| | Do not accept medical assistance (e.g., Medicaid/Medicare) | Yes | 11 (31.4) | 8 (30.8) | 7 (35.0) | 12 (30.0) | 5 (21.7) | 14 (36.8) | 8 (28.6) | 11 (33.3) |
| | | No | 24 (68.6) | 18 (69.2) | 13 (65.0) | 28 (70.0) | 18 (78.3) | 24 (63.2) | 20 (71.4) | 22 (66.7) |
| | | p | 0.956 | | 0.695 | | 0.217 | | 0.689 | |
| | Requires a multidisciplinary approach | Yes | 5 (14.3) | 6 (23.1) | 5 (25.0) | 6 (15.0) | 3 (13.0) | 8 (21.1) | 6 (21.4) | 5 (15.2) |
| | | No | 30 (85.7) | 20 (76.9) | 15 (75.0) | 34 (85.0) | 20 (87.0) | 30 (78.9) | 22 (78.6) | 28 (84.8) |
| | | p | 0.504 | | 0.481 | | 0.511 | | 0.525 | |
| | Dental office not properly equipped | Yes | 7 (20.0) | 4 (15.4) | 4 (20.0) | 7 (17.5) | 4 (17.4) | 7 (18.4) | 4 (14.3) | 7 (21.2) |
| | | No | 28 (80.0) | 22 (84.6) | 16 (80.0) | 33 (82.5) | 19 (82.6) | 31 (81.6) | 24 (85.7) | 26 (78.8) |
| | | p | 0.745 | | 1.00 | | 1.00 | | 0.483 | |
| | Other | Yes | 7 (20.0) | 7 (26.9) | 6 (30.0) | 8 (20.0) | 7 (30.4) | 7 (18.4) | 7 (25.0) | 7 (21.2) |
| | | No | 28 (80.0) | 19 (73.1) | 14 (70.0) | 32 (80.0) | 16 (69.6) | 31 (81.6) | 21 (75.0) | 26 (78.8) |
| | | p | 0.525 | | 0.519 | | 0.280 | | 0.726 | |

**Note:**
[†],* Data are presented as frequencies (column percentages).

**Table 4 Associations between reported dentist-level variables and perceptions of the need for dental educational materials for the LS community[†].**

| | | Dental specialty | | Years of dental experience | | Number of days per week currently seeing patients | | Number of patients currently seen per week | |
|---|---|---|---|---|---|---|---|---|---|
| | | General dentist | Specialist | 0–20 | 21+ | 0–3 | 4+ | 0–40 | 41+ |
| Feel there is a need for more educational materials to help patients with Lowe syndrome in the dental setting | Strongly agree | 17 (50.0) | 8 (33.3) | 7 (36.8) | 18 (47.4) | 9 (45.0) | 16 (42.1) | 12 (46.2) | 13 (40.6) |
| | Agree | 9 (26.5) | 13 (54.2) | 10 (52.6) | 11 (28.9) | 5 (25.0) | 17 (44.7) | 7 (26.9) | 15 (46.9) |
| | Neutral | 8 (23.5) | 3 (12.5) | 2 (10.5) | 9 (23.7) | 6 (30.0) | 5 (13.2) | 7 (26.9) | 4 (12.5) |
| | Disagree | 0 (0.0) | 0 (0.0) | 0 (0.0) | 0 (0.0) | 0 (0.0) | 0 (0.0) | 0 (0.0) | 0 (0.0) |
| | Strongly disagree | 0 (0.0) | 0 (0.0) | 0 (0.0) | 0 (0.0) | 0 (0.0) | 0 (0.0) | 0 (0.0) | 0 (0.0) |
| | p | 0.615 | | 0.942 | | 0.627 | | 0.794 | |

**Note:**
[†] Data are presented as frequencies (column percentages).

who had 0–20 years of dental experience were significantly more likely, compared with subjects who had 21+ years of dental experience, to report feeling that media channels (such as TikTok, Instagram, and YouTube) would be helpful for individuals with LS ($p = 0.018$). All other associations were not statistically significant.
Table 5 Associations between reported dentist-level variables and perceptions of the need for dental educational materials for the LS community[†].

| Feel the following types of educational materials would be helpful for individuals with Lowe syndrome | Website/Mobile application | Yes | 23 (71.9) | 14 (58.3) | 11 (64.7) | 25 (65.8) | 12 (60.0) | 25 (69.4) | 13 (54.2) | 24 (75.0) |
|---|---|---|---|---|---|---|---|---|---|---|
| | | No | 9 (28.1) | 10 (41.7) | 6 (35.3) | 13 (34.2) | 8 (40.0) | 11 (30.6) | 11 (45.8) | 8 (25.0) |
| | | p | 0.290 | | 0.938 | | 0.474 | | 0.103 | |
| | Introductory dental video | Yes | 22 (68.8) | 19 (79.2) | 14 (82.4) | 26 (68.4) | 15 (75.0) | 26 (72.2) | 15 (62.5) | 26 (81.3) |
| | | No | 10 (31.3) | 5 (20.8) | 3 (17.6) | 12 (31.6) | 5 (25.0) | 10 (27.8) | 9 (37.5) | 6 (18.8) |
| | | p | 0.384 | | 0.344 | | 0.822 | | 0.117 | |
| | Pamphlets with oral hygiene instructions | Yes | 23 (71.9) | 17 (70.8) | 13 (76.5) | 26 (68.4) | 16 (80.0) | 24 (66.7) | 18 (75.0) | 22 (68.8) |
| | | No | 9 (28.1) | 7 (29.2) | 4 (23.5) | 12 (31.6) | 4 (20.0) | 12 (33.3) | 6 (25.0) | 10 (31.3) |
| | | p | 0.932 | | 0.750 | | 0.290 | | 0.608 | |
| | Media channels (e.g., TikTok/ Instagram/ YouTube) | Yes | 8 (25.0) | 9 (37.5) | 9 (52.9) | 8 (21.1) | 5 (25.0) | 12 (33.3) | 4 (16.7) | 13 (40.6) |
| | | No | 24 (75.0) | 15 (62.5) | 8 (47.1) | 30 (78.9) | 15 (75.0) | 24 (66.7) | 20 (83.3) | 19 (59.4) |
| | | p | 0.314 | | 0.018* | | 0.516 | | 0.054 | |
| | Podcasts featuring dental experts | Yes | 8 (25.0) | 11 (45.8) | 6 (35.3) | 13 (34.2) | 9 (45.0) | 10 (27.8) | 9 (37.5) | 10 (31.3) |
| | | No | 24 (75.0) | 13 (54.2) | 11 (64.7) | 25 (65.8) | 11 (55.0) | 26 (72.2) | 15 (62.5) | 22 (68.8) |
| | | p | 0.103 | | 0.938 | | 0.192 | | 0.625 | |
| | Focus groups | Yes | 9 (28.1) | 7 (29.2) | 5 (29.4) | 11 (28.9) | 6 (30.0) | 10 (27.8) | 5 (20.8) | 11 (34.4) |
| | | No | 23 (71.9) | 17 (70.8) | 12 (70.6) | 27 (71.1) | 14 (70.0) | 26 (72.2) | 19 (79.2) | 21 (65.6) |
| | | p | 0.932 | | 1.00 | | 0.860 | | 0.267 | |

Note:
[†,*] Data are presented as frequencies (column percentages).

## DISCUSSION

Given the substantial oral health problems frequently experienced by individuals with LS (*Knight & Scientific Advisory Board, 2010*), access to dental care is crucial for this community. The current research on the experience and perceptions of dentists regarding LS serves as a complement to previous surveying of parents/guardians (*Lowenstein et al., 2023*), thereby providing a fuller picture of the barriers to dental care faced by individuals with this debilitating disorder. For instance, our finding that three-quarters of dentists reported at least one reason why they might not be able to treat an individual with LS may partially explain the results of a prior study in which only 13% of parents/guardians of individuals with LS reported it was "very easy" to locate a dentist for the individual with LS (*Lowenstein et al., 2023*). Interestingly, although a "lack of experience treating children with special needs" was among our most commonly reported barriers to providing dental care, no respondents with 0–20 years of dental experience reported this barrier. This result suggests that dental schools may have placed greater emphasis on special care in dentistry within their curricula in recent years, and/or that recent graduates may be seeking opportunities to gain experience in this domain. Therefore, LS modules should be added to dentistry curricula. In doing so, awareness of LS will increase, and the goal would be to improve the knowledge base. If the knowledge base is increased, then policy change can occur. In the past, dentists relied on paediatric-oriented skills they acquired during their

undergraduate training without any adaptation while treating patients with IDDs, which only exacerbates the barriers to care for this population (*Phadraig et al., 2020*). Although in recent years, schools have shifted to requiring dental graduates to provide treatment using patient support techniques or non-pharmacological/non-physical techniques for patients requiring special care (*Phadraig et al., 2020*).

Nevertheless, only 5% of dentists reported having ever treated a patient with LS. While the latter finding can largely be attributed to the rarity of the disorder, it also reflects the difficulty that parents and caregivers may encounter in finding a dentist who has experience with LS. In fact, in the current research, more dentists reported that they had not previously heard of LS than those who reported having treated a patient with the condition. Such a finding illuminates the need for greater awareness of LS and its effects on oral health among dentists. This begs the question of how do we increase the awareness of not just LS, but also other IDD's and the needs of special care dentistry? The American Dental Education Association (ADEA) in 2006 adopted a resolution to include didactic instruction and clinical experiences treating people with special needs (*Faulks et al., 2012*). However, the quality, method, and content of teaching varies widely amongst all dental schools as there is no universal curriculum to follow and most often this type of module is linked together with paediatric dentistry (*Faulks et al., 2012*). Multiple studies have concluded how such training is inadequate and graduating dentists do not have enough exposure to conditions such as LS. This is where dental education materials can make the difference. We can bridge that gap in knowledge and awareness amongst dentists and special care dentistry where institutions have failed or may not have the experts required to teach this material. Over the last decade, only one article has detailed the dental needs and conditions of individuals with Lowe syndrome.

Another compelling finding was that approximately four-fifths of respondents agreed or strongly agreed that there is a need for more educational materials to help patients with LS in the dental setting. Future research could focus on developing and testing the types of materials that were most frequently identified as helpful (an introductory dental video, pamphlets with oral hygiene instructions, and a website or mobile application). It is noteworthy that although media channels were not among the materials most commonly identified as helpful, the significantly greater endorsement of such channels by dentists with 0–20 years of dental experience may reflect generational changes in preference for how dental information is obtained or disseminated. We emphasize that all dental educational materials customized for the LS community, regardless of their type, should be vetted by parents, caregivers, and other stakeholders at each stage of the development and testing process.

The need for these materials now is more than ever. A case study in Brazil in 2011 documented the first orthodontic treatment in a patient with LS (*Ruellas et al., 2011*). The team had to simplify the mechanical procedures due to the patient's condition but were ultimately able to provide the patient with improved occlusion, esthetics, and quality of life (*Ruellas et al., 2011*). Their biggest challenge was cooperation in the chair which is where dental education materials could have served as a guide for the dentists. This is one of the few cases where dentists were able to achieve satisfactory results, but that does not have to

be the norm. The creation of these materials will allow every patient with LS to receive proper care without drastic changes to the treatment plan.

One limitation of this research is the potential for self-selection bias, *i.e.*, the potential lack of representativeness of the convenience sample due to the fact that each prospective subject decided for themselves whether to participate (*Bethlehem, 2010*; *Moore & McCabe, 1999*). Additionally, the response rate was low (approximately 1.4% based on the estimated 5,500 email addresses on Tufts University School of Dental Medicine's listserv), which is common in dental survey research (*Magnuson et al., 2020*) but may exacerbate the potential for unrepresentativeness. Additionally, self-report surveys may be prone to social desirability bias, in which subjects answer questions to convey a greater level of socially acceptable beliefs or behaviors than is accurate (*Bispo Júnior, 2022*). However, some authors have expressed that self-report surveys may exhibit a higher level of validity than is typically perceived (*Chan, 2009*). We also note that our evaluation of associations was exploratory, and findings should be confirmed in replication studies. Finally, Cronbach's alpha, which can be a useful statistic in assessing the internal consistency/reliability of items on a questionnaire, was not pertinent to our survey because it was not designed or used to measure a single latent construct.

## CONCLUSIONS

Most dentists do not have experience in treating individuals with LS and perceive at least one reason why they might not be able to treat an individual with LS. Therefore, substantial barriers to dental care exist for individuals with LS. Developing tailored educational materials about the dental setting, such as an introductory dental video, pamphlets with oral hygiene instructions, and a website or mobile application should be developed for the LS community. The value of dental educational materials may serve as a means to reduce inequalities in special oral care.

### Funding
The authors received no funding for this work.

### Competing Interests
Carlos Fernando Mourão is an Academic Editor for PeerJ. The authors declare no other competing interests, financial or non-financial, professional or personal, related to this manuscript.

### Author Contributions
- Adam Lowenstein conceived and designed the experiments, performed the experiments, analyzed the data, prepared figures and/or tables, authored or reviewed drafts of the article, and approved the final draft.
- Matthew Finkelman conceived and designed the experiments, analyzed the data, prepared figures and/or tables, authored or reviewed drafts of the article, and approved the final draft.

- Jay Dalal performed the experiments, prepared figures and/or tables, and approved the final draft.
- Crystal Smith performed the experiments, prepared figures and/or tables, and approved the final draft.
- Glory Ogunyinka performed the experiments, prepared figures and/or tables, and approved the final draft.
- David Tesini conceived and designed the experiments, analyzed the data, authored or reviewed drafts of the article, and approved the final draft.
- Carlos Fernando Mourão conceived and designed the experiments, analyzed the data, prepared figures and/or tables, authored or reviewed drafts of the article, and approved the final draft.

### Human Ethics

The following information was supplied relating to ethical approvals (*i.e.*, approving body and any reference numbers):

Tufts University School of Dental Medicine. Approval number: 00004167.

### Data Availability

The raw data are available in the Supplemental Files.

### Supplemental Information

Supplemental information for this article can be found online at http://dx.doi.org/10.7717/peerj.20174#supplemental-information.

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
