# Peer review of "Barriers to care and the need for dental educational materials for the Lowe syndrome community: a survey of dentists"

_PeerJ, doi:10.7717/peerj.20174_

## Round 0.1 · original submission · Major Revisions

Please respond to the comments from Reviewer 1 and (especially) Reviewer 5.

Reviewer 1 ·

Basic reporting

The author needs to provide details of the dentistry curriculum followed at the university to understand whether the dentists were trained to treat LS patients during their training.

Experimental design

The methods did not contain information about the sampling technique used.

Please provide details on how the questionnaire was developed.

The author needs to mention the reason for selecting a dentist from a single dental school for this survey.

Need to mention the inclusion and exclusion criteria of participants.

Validity of the findings

Please report the response rate.

How many participants were involved in the questionnaire pre-testing, and were they part of the study or not?

Why have you not reported the Cronbach's alpha of the questionnaire?

Table 5 is difficult to understand. Redo table 5.

Additional comments

Send my above comments to the author for corrections.

·

Basic reporting

-

Experimental design

-

Validity of the findings

-

Reviewer 3 ·

Basic reporting

The manuscript is well-written and represents the study as a whole. The background of the study is adequate, with relevant citations. The tables are informative, including the p-values. However, no hypothesis is mentioned.

Experimental design

The study is well-designed within the scope of the journal. The methods and anonymous participation are well-presented and relevant to the study. The study demonstrates accurate methods of analysis and adheres to ethical standards.

Validity of the findings

The validity of the findings is provided in the supplemental files. The study data is well-kept confidentially, and the presentation is organized. The summary of the study and its findings is well-explained, highlighting the benefits of the study to the community in understanding Lowe Syndrome.

Additional comments

Understanding Lowe Syndrome is crucial in dentistry because of the potential dental, systemic, and developmental challenges it presents. Dentists need to be aware of these issues to provide effective, tailored care and contribute to a multidisciplinary approach to managing the condition.

·

Basic reporting

Background: Write a few statements about the subject. This objective is not background
Keywords: add keywords
References should be numbered in the reference section
Some of the references are too old and need to be updated

Experimental design

well represented

Validity of the findings

Results are clear and concise.

Reviewer 5 ·

Basic reporting

The manuscript is generally clearly structured, but the following minor English language corrections should be noted.

The explanations in parentheses in the sentence between lines 41-43 seem a bit confusing. It can be corrected as follows for a clearer explanation. "Lowe syndrome (LS), also known as oculocerebrorenal syndrome (OCRL), is a rare disorder..."

In the sentence between lines 64-67, a clearer expression could be preferred instead of the expression "Appointments in which general anesthesia was used": "A 1991 study reported that subjects with the disorder were more likely than the general population to experience many of the problems listed above, including misalignment, extractions, and the need for general anesthesia during dental procedures."
"Educational materials about the dental setting, such as an introductory dental video, pamphlets with oral hygiene instructions, and a website or mobile application, should be developed for the LS community." Such statements are repeated more than once in the article. Unnecessary repetitions can be avoided.
The article is generally well written and clear, but the minor language and wording issues noted above could be improved to make it more professional. It is particularly important to avoid repetition and use consistent terminology.

The manuscript is well structured with literature references and background review to provide context.
Study Strengths: A wealth of literature is referenced to assess the clinical and dental manifestations of Lowe syndrome. Previous survey studies (Tesini 2023, McSpadden et al. 1991) and case reports (Ruellas et al. 2011) are cited to support barriers to accessing dental care. The importance of educational materials is particularly emphasized, and the 2006 ADEA (American Dental Education Association) resolution on special care dentistry education is cited. The introduction clearly outlines the systemic and dental manifestations of LS (e.g., “tooth alignment problems, need for general anesthesia”). Highlighting the knowledge gap: The difficulties experienced by individuals with LS in accessing dental care are supported by previous parent surveys (Tesini 2023) and are linked to the purpose of the article.

Weaknesses of the study: Since the subject of the study is a rare disease, the literature focuses on the year 2023 at most. Since the number of studies is low, there are experimental studies rather than clinical studies. Global context is lacking: The study focuses on dentists in the USA (Tufts University alumni). A brief reference to the prevalence of LS worldwide and dental care policies may be made.

The article is a self-contained study that provides results consistent with its hypotheses. However, the following points should be noted:
Interpretation of negative findings: Non-significant relationships should also be discussed.
Sample limits: It should be more clearly emphasized that the results are specific to Tufts graduates.
These changes will increase the transparency and reliability of the study.

Experimental design

The article fills an important knowledge gap (dental care barriers and dentist perspective in LS) by clearly defining the research question. However, the following improvements could be made:
Clearly stating the hypotheses (should be included in the Introduction).

Highlighting the literature gap quantitatively (e.g., “Only 2 studies in the last 10 years…”).
More elaboration of policy/clinical recommendations in the discussion (e.g., “LS modules should be added to dentistry curricula”).

These changes would further strengthen the unique contribution and impact of the study.
The study meets basic technical and ethical standards, but the following improvements could be made:
Single-Center Sample: Limited to Tufts graduates, may not be representative of all dentists in the United States. Medicaid acceptance rates may vary regionally.

Pediatric Dentists in Minority: While individuals with LS are commonly treated by pediatric dentists, the sample included only 8 pediatric dentists (n=73) (Table 1). This weakens specialty-based analyses.

Validity of the findings

Emphasis on Novelty: The introduction or discussion should state more strongly that LS is rare in the dental literature and that this is the first time the dentist's perspective has been examined.
Clinical Implication: One or two concrete suggestions should be added on how the results will be reflected in dental education and health policies.

Conclusion: The article is a coherent study that meets technical and ethical standards and presents results consistent with its hypotheses. Its contribution can be emphasized more clearly with minor improvements.

Additional comments

Your study provides valuable insights into dental care barriers for Lowe Syndrome patients. While the methodology is sound, please clarify data availability (e.g., full survey questions) and briefly address sample limitations in the discussion. Minor language edits would enhance readability. Overall, this is a meaningful contribution to a rare but critical topic.

---

## Round 0.2 · accepted · Accept

Thank you for revising your manuscript to address the reviewers' concerns. I am satisfied that your revisions address the comments of our various reviewers. The manuscript is now ready for publication.

Reviewer 3 ·

Basic reporting

The research background is well explained.

Experimental design

The methods are well written.

Validity of the findings

The findings are meaningful and beneficial to the community.